# Genome-Wide Analysis of ATP Binding Cassette (ABC) Transporters in Peach (*Prunus persica*) and Identification of a Gene *PpABCC1* Involved in Anthocyanin Accumulation

**DOI:** 10.3390/ijms24031931

**Published:** 2023-01-18

**Authors:** Cherono Sylvia, Juanli Sun, Yuanqiang Zhang, Charmaine Ntini, Collins Ogutu, Yun Zhao, Yuepeng Han

**Affiliations:** 1CAS Key Laboratory of Plant Germplasm Enhancement and Specialty Agriculture, Wuhan Botanical Garden, The Innovative Academy of Seed Design, Chinese Academy of Sciences, Wuhan 430074, China; 2Hubei Hongshan Laboratory, Wuhan 430070, China; 3University of Chinese Academy of Sciences, Beijing 100049, China; 4Sino-African Joint Research Center, Chinese Academy of Sciences, Wuhan 430074, China

**Keywords:** anthocyanin, ABC transporter, gene expression, *PpABCC1*, *Prunus persica*

## Abstract

The ATP-binding cassette (ABC) transporter family is a large and diverse protein superfamily that plays various roles in plant growth and development. Although the ABC transporters are known to aid in the transport of a wide range of substrates across biological membranes, their role in anthocyanin transport remains elusive. In this study, we identified a total of 132 putative *ABC* genes in the peach genome, and they were phylogenetically classified into eight subfamilies. Variations in spatial and temporal gene expression levels resulted in differential expression patterns of *PpABC* family members in various tissues of peach. *PpABCC1* was identified as the most likely candidate gene essential for anthocyanin accumulation in peach. Transient overexpression of *PpABCC1* caused a significant increase in anthocyanin accumulation in tobacco leaves and peach fruit, whereas virus-induced gene silencing of *PpABCC1* in the blood-fleshed peach resulted in a significant decrease in anthocyanin accumulation. The *PpABCC1* promoter contained an MYB binding *cis*-element, and it could be activated by anthocyanin-activator PpMYB10.1 based on yeast one-hybrid and dual luciferase assays. Thus, it seems that *PpABCC1* plays a crucial role in anthocyanin accumulation in peach. Our results provide a new insight into the vacuolar transport of anthocyanins in peach.

## 1. Introduction

ATP binding cassette (ABC) proteins constitute one of the largest diverse superfamilies of transporters in cells of all species [1,2]. These transporters function in the translocation of a wide range of substrates across membranes through ATP binding and hydrolysis and can be characterized as either importers or exporters depending on the direction of substrate transport [3]. ABC transporters consist of four structural domains: two transmembrane domains (TMDs) and two cytoplasmic nucleotide-binding domains (NBDs) [4,5]. TMDs are composed of several membrane-spanning α-helices that regulate substrate recognition and specificity [6]. NBDs consist of highly conserved motifs, including Walker A, Walker B, the ABC signature, the H loop and the Q loop [3,7]. 

Plant ABC transporter proteins are classified into three types based on their domain structure: full-sized transporters composed of two TMD and two NBD domains, half-sized transporters containing a single TMD and NBD domain, and incomplete or divergent transporters characterized by two NBD domains but lack of TMD [3,8,9]. ABC transporters in plants are clustered into eight subfamilies, ABCA-ABCG and ABCI, based on domain structural organization and functional characterization. The ABCH subfamily is exclusively present in the animal genome [3,10]. 

ABCA subfamily comprises one full-sized transporter and several half-sized transporters (Appendix A). The transmembrane transporter is involved in different functions in animals, including lipid metabolism, cholesterol modulation and lipoprotein transport. However, its physiological role in plants still remains unclear [11,12]. ABCB subfamily is classified into full-sized transporters known as multidrug resistance protein (MDR) or P-glycoprotein (PGP) and half-sized transporters that are categorized into three groups, transporter associated with antigen processing (TAP), lipid A-like exporters, putative (LLP) and ABC transporters of the mitochondria (ATP). These proteins are mainly involved in the transport of organic compounds across the plasma membrane [13,14]. ABCD subfamily, also known as peroxisomal membrane proteins (PMPs), consists of both full and half-sized transporters and functions in the peroxisomal transport of fatty acids across the plant membrane [15,16,17]. Soluble plant proteins are located in the ABCE, ABCF and ABCI subfamilies and comprise two NBDs but lack the TMD domain. These proteins perform different functions in plants, such as RNA interference (RNAi), gene expression regulation and plastid lipid synthesis [10,18]. ABCG constitutes the largest subfamily with a reverse domain organization (NBD-TMD). It comprises both full- and half-sized transporters that are known as pleiotropic drug resistance (PDR) and white-brown complex (WBC), respectively [3]. The ABCG-type transporters are localized in the plasma membrane and function in the transport of phytohormones and pathogen defense [10,19].

ABCC subfamily is full-sized transporters that are also known as multidrug resistance-associated proteins (MRP) due to their role in vacuolar sequestration of glutathione (GSH) and glucuronide-conjugates [3]. ABCC subfamily has a similar domain organization to MDR proteins, but they differ in the N-terminal region where a hydrophobic N-terminal extension (NTE or TMD0) is present in the MRP proteins but absent in the MDR proteins [20]. Additionally, the ABCC subfamily has been shown to function in stomatal regulation, phytate transport, vacuolar folate transport, chlorophyll catabolite transport, heavy metal and cadmium stress tolerance [21].

Recent studies suggest that the ABCC subfamily plays an important role in the accumulation of anthocyanin in plants. In cereals, the maize *ZmMrp3* gene and the rice *OsMRP*15 gene have both been reported to transport anthocyanins and flavonoids into the vacuole [22,23]. In grapes, the *VvABCC1* gene acts as an anthocyanidin 3-*O*-glucosides transporter and is GSH-dependent without the formation of anthocyanin-GSH conjugates [24]. Transcriptome analysis of black and white spine grapes (*Vitis davidii*) revealed that both ABCC1 and ABCC2 were essential for anthocyanin accumulation, with higher expression observed in black berry grapes [25]. In *Arabidopsis thaliana*, the *AtABCC2* gene has been reported to be involved in the transport of anthocyanins and various flavonoid compounds in the vegetative tissues [26]. *AtABCC1* and *AtABCC14* have recently been reported to aid in the uptake of acylated anthocyanins in vitro and are not dependent on glutathione conjugation as previously seen with other ABCC transporters for anthocyanidin 3-O-monoglucosides [27].

Despite recent advances in genome-wide analysis of the ABC transporters in various plant species such as *Arabidopsis* [16], rice [18], maize [11], pineapple [28], pepper [29], soybean [30], rapeseed [31,32], tomato [1], strawberry [8] and grape [24], little is known about the ABC transporters in peach. Peach (*Prunus persica* L. (Batsch)) belongs to the Rosaceae family and is an important economic fruit tree cultivated worldwide. Due to its relatively small genome size (~230 Mb), the peach is deemed a model fruit species for comparative and functional genomics in deciduous woody perennial trees [33]. Anthocyanin pigmentation is an essential factor determining peach fruit quality. Previous studies have reported that multidrug and toxic extrusion (MATE), glutathione S-transferases (GSTs) and ABC proteins mediate vacuolar sequestration of anthocyanins in perennial fruit crops [22,34,35,36,37]. Due to the significance of anthocyanin accumulation in determining fruit quality in peaches, we investigated here the role of ABC transporters in anthocyanin accumulation. In this work, we identified the ABC members in peach and found a candidate ABC transporter involved in anthocyanin vacuolar sequestration in peach fruit. Our results will provide valuable information that contributes to a better understanding of the ABC family and the molecular mechanisms underlying anthocyanin accumulation in peach fruit. 

## 2. Results

### 2.1. Identification of ABC Transporters in Peach

A total of 132 ABC genes were identified in the peach genome. Detailed information on the chemical characterization, including the length, Mw, theoretical pI, total number of negatively/positively charged amino acids, GRAVY, aliphatic and instability index of the identified 132 ABC gene family, were analyzed (Appendix A). The protein length of the 132 PpABC proteins ranged from 71 (Prupe.1G250300) amino acids (aa) to 2524 (Prupe.1G103900) aa with an average of 949.61 aa. The predicted Mws of the ABC proteins ranged from 8188.53 (Prupe.1G250300) to 271552.51 (Prupe.1G103900) kDA with an average of 105449.96 kDA, while the pIs ranged from 5.05 (Prupe.I000500) to 10.48 (Prupe.5G155100) with an average of 7.94. The number of negatively/positively charged amino acids ranged from 6 (Prupe.5G155100) to 256 (Prupe.1G103900) and from 8 (Prupe.1G250300) to 242 (Prupe.1G103900), respectively. A total of 44 PpABC proteins were predicted to be hydrophilic proteins due to their relatively low hydropathy score of GRAVY (value < 0). The aliphatic index of the PpABC proteins ranged from 69.96 (Prupe.3G046700) to 118 (Prupe.1G148500). Most of the PpABC proteins (89) were deemed stable due to their instability index being less than 40 (Appendix A). Subcellular localization predictions showed that the majority of the PpABC proteins (103) were located in the plasma membrane, while 11, 8, 5 and 4 were located in the cytoplasmic membrane, nucleus, mitochondria and chloroplast, respectively. Only one ABC (Prupe.5G155200) was predicted to be located in the extracellular space. Several PpABC proteins (11) were found located in multiple positions (Appendix A).

### 2.2. Phylogenetic Analysis of PpABCs

To explore the evolutionary relationship of the ABC transporters, a phylogenetic tree was constructed using the amino acid sequences of PpABCs (Figure 1). The 132 *PpABC* genes were divided into eight subfamilies (A–G and I). In our analysis, ABCG had the largest number of members in all subfamilies. ABCB was the second largest subgroup comprising 29 members, followed by the ABCC, which had 26 members. Subfamily ABCI had 13 members, while subfamilies ABCA, ABCE, ABCF and ABCD had 9, 5, 3 and 2 members, respectively (Appendix A).

The evolutionary history and relationship of the ABC gene family in plants were analyzed by comparing the ABC distribution in twelve different species (Appendix A). The number of each subfamily showed great variations among plant species. In general, we discovered that ABCA, ABCB, ABCC, ABCG and ABCI appeared to have expanded more than the other groups, while ABCD, ABCE and ABCF constituted the least number of members in all the studied species. In peach, the numbers of the ABCB, ABCC, ABCG and ABCI subfamilies were much higher than those of the other subfamilies. 

### 2.3. Gene Structure and Conserved Motifs of PpABC Gene Family

Since structural analysis is important in understanding the functional and evolutionary relationships between homologous genes, we examined the exon–intron structure of the *PpABC* gene family (Appendix A). The highest number of exons (forty) was observed in the ABCA gene *Prupe.7G058100*. By contrast, the least number of exons was identified in five ABCG genes, *Prupe.4G168100*, *Prupe.4G272400*, *Prupe.4G266200*, *Prupe.2G303100* and *Prupe.2G224800*, and one ABCC gene *Prupe.7G259900*, all of which had only one exon and lacked introns. 

The structural diversity of *PpABC* genes was further investigated using the conserved motif analysis (Appendix A). A total of fifteen conserved motifs were identified in the PpABC proteins. Closely related ABC members showed similarity in motif alignment and position, suggesting that ABC proteins clustered together in the same group might perform similar biological functions. Motif 9 was found to be unique to the ABCG group, while motifs 2 and 3 were present in most of the eight subfamilies displaying the structural divergence of the PpABCs proteins.

### 2.4. Chromosomal Distribution and Collinearity Analysis of PpABC Genes

Chromosomal distribution analysis indicated the uneven distribution of 132 ABC genes on eight chromosomes in the peach genome, with three *PpABC* genes (*Prupe.I000500*, *Prupe.I000600* and *Prupe.I000700*) mapped under unassembled scaffolds (Figure 2, Appendix A). In the peach genome, chromosomes (Chr) 1 and Chr3 contained the most ABC genes (25 members each), followed by 16 genes located on Chr4 and 15 genes located on Chr6. The hierarchical chromosomal distribution of the *PpABC* genes in different subfamilies was observed. The *PpABC* genes of group G, with the most members (43), were randomly distributed across the eight chromosomes. Groups A (9 members), B (29 members), C (26 members), D (2 members), E (5 members), F (3 members) and I (13 members) were located on 3, 7, 7, 2, 2, 3 and 6 chromosomes, respectively. To further understand the evolutionary and collinearity relationship of the *PpABC* gene family, we performed the syntenic collinearity analysis of the ABC family in peach. A total of 3, 4 and 2 collinear gene pairs were identified in ABCC, ABCG and ABCI subgroups, respectively (Appendix A). These results suggested that segmental duplications contributed to the expansion of *PpABC* genes.

### 2.5. Expression Pattern of PpABC Genes in Different Tissues of Peach 

In order to further explore the expression pattern of *PpABC* genes in peach, transcription levels of *PpABC* genes in different tissues such as leaf, stem, root and callus were analyzed. A total of 119, 109, 105 and 100 *PpABC* genes were expressed in the leaf, stem, root and callus, respectively (Figure 3, Appendix A). A total of 123 *PpABC* genes (accounting for 93.2% of total *PpABCs*) were expressed in all analyzed tissues. Among these, sixteen *PpABC* genes showed high expression levels (FPKM > 15) in all tested tissues. In addition, some *PpABC* genes presented tissues-specific expression, with 42 *PpABC* genes (34% of 123 expressed *PpABC* genes), 29 *PpABC* genes (24%), 27 *PpABC* genes (22%) and 22 *PpABC* genes (18%) specifically expressed in leaf, callus, root and stem, respectively. The expression of nine *PpABC* genes was not detected in these tissues. Overall, these results provide a baseline for the identification of various functional genes in peach.

The role of *PpABC* genes in regulating fruit development was also investigated. The expression profiles of *PpABC* genes during fruit development were estimated using fruit samples of two peach cultivars, ‘Dahongpao’ and ‘Jinxiang’, which were collected at the green stage (S1) and color break stage (S2). A heatmap representing the transcript expressions of the detected *PpABC* genes was constructed (Figure 4, Appendix A). A total of 102 *PpABC* genes were expressed in fruit, and 67 had an expression level over 1 (FPKM) at both developmental stages. A total of 23 *PpABC* genes all showed higher expression (FPKM > 15) in the S2 stage of both cultivars. Among the 23 *PpABC* genes, 2, 4, 6, 1, 3, 3, 1 and 3 belonged to subgroups A, B, C, D, E, F, G and I, respectively. Six *PpABCC* members had higher expression profiles in fruits of ‘Dahongpao’ at the S2 stage. Given that the *ABCC* members are reported to be related to flavonoid biosynthesis [38], these six *PpABCC* genes may play critical roles in anthocyanin accumulation. 

### 2.6. Analysis of Cis-Elements in the PpABCC Promoter

The putative *cis*-acting elements in the promoter region were identified for all the *PpABC* genes except *Prupe.I000500* due to the lack of its promoter sequence in the database. The *cis*-acting regulatory elements of *PpABCC* subfamily members were classified into three groups based on their functional annotation: plant growth and development, phytohormone responsive, and abiotic and biotic stress (Figure 5). The majority of the elements (27) in the *PpABCC* subgroup members were related to abiotic and biotic stress, suggesting their important roles in stress response. However, the number of stress-related regulatory elements varied between the *PpABCC* subgroup members. The *cis*-elements related to abiotic and biotic stress were abundant, including light-responsiveness elements (G-Box) and low-temperature-responsive elements (LTR). The second group is the phytohormone response group, which consisted of twelve elements, with ABRE, ERE, TGACG, and CGTCA-motifs being widely distributed among *PpABCC* subfamily members. They are involved in abscisic acid (ABA), ethylene, and methyl jasmonate (MeJA) responsiveness in plants, respectively. The third group was related to plant growth and development and had nine elements, such as the GCN4-motif, O_2_ site, and MBSI, involved in endosperm expression, zein metabolism, and flavonoid biosynthetic gene regulation, respectively. In addition, an MYB binding site element (MBSI) was found in the promoter region of *PpABCC1* (*Prupe.4G088500*). These results suggested that *ABCC* genes play important roles in a variety of developmental and physiological processes, including flavonoid biosynthesis in peach. 

### 2.7. Identification of PpABCC Genes Regulating Anthocyanin Accumulation in Peach

To identify the ABCC genes involved in anthocyanin accumulation in peach, a phylogenetic tree was constructed based on *Arabidopsis* C-subgroup and *PpABCC* members (Appendix A). Among the 26 candidates ABCC members in peach, *PpABCC1* was found to be clustered in the same group with the *Arabidopsis* gene *AtABCC2,* which was responsible for flavonoid transport in *Arabidopsis* [26]. The *PpABCC1* gene was highly expressed in fruits of blood peach ‘Dahongpao’ at the S2 stage when anthocyanin accumulation occurred but had very weak expression in fruits of yellow peach ‘Jinxiang’ at both S1 and S2 stages (Figure 4). This suggested that the *PpABCC1* gene could be involved in fruit anthocyanin accumulation in peach. 

To verify this hypothesis, we carried out a correlation analysis between anthocyanin content and the *PpABCC1* transcript abundance in mature fruits of twenty peach cultivars (Figure 6A, Appendix A). Both the *PpABCC1* expression and anthocyanin content varied among cultivars, with the highest expression of *PpABCC1* observed in the ‘97-8-1′ cultivar that contained the highest content of anthocyanins. Linear regression analysis showed a strong positive correlation of the *PpABCC1* expression with anthocyanin content among different cultivars (r = 0.974, *p* < 0.01). In addition, the subcellular localization assay showed that PpABCC1 was localized in the nuclei and the tonoplast (Figure 6B). Taken together, these results suggested that *PpABCC1* is the most likely candidate gene involved in anthocyanin accumulation in peach. 

### 2.8. Functional Analysis of PpABCC1 Using Transient Transformation Assay

In order to validate the role of *PpABCC1* in anthocyanin coloration in vivo, its transient overexpression was performed in tobacco leaves and peach fruits. Tobacco leaves were infiltrated with the combination of *PpMYB10.1*, *PpbHLH3* and *PpABCC1,* and the infiltration of *PpMYB10.1* + *PpbHLH3* was used as the control. Five days after injection, deep purple color patches were observed around the sites infiltrated with either *PpMYB10.1*/*PpbHLH3*/*PpABCC1* or *PpMYB10.1*/*PpbHLH3* (Figure 7A). However, leaf tissues around the site infiltrated with *PpMYB10.1*/*PpbHLH3*/*PpABCC1* had a stronger pigmentation compared to those infiltrated with *PpMYB10.1*/*PpbHLH3* which only showed a slight change in coloration. Anthocyanin content in leave tissues infiltrated with *PpMYB10.1*/*PpbHLH3*/*PpABCC1* was 1.6-fold higher than that in leave tissues infiltrated with *PpMYB10.1*/*PpbHLH3* (Figure 7A). *PpABCC1*-overexpression significantly increased the expression of *NtANS* by 1.8-fold, which was well consistent with the increased anthocyanin accumulation in the tobacco leaves (Figure 7A).

Similarly, intense red color patches appeared one week after infiltration in both the flesh and peel infiltrated with *PpMYB10.1*/*PpbHLH3*/*PpABCC1* compared to these infiltrated with *PpMYB10.1*/*PpbHLH3* (Figure 7B,C). Anthocyanin content in the flesh around the sites infiltrated with *PpMYB10.1*/*PpbHLH3*/*PpABCC1* showed 1.5-fold higher as compared to that in the flesh around the sites infiltrated with *PpMYB10.1*/*PpbHLH3*. *PpABCC1*-overexpression significantly increased the expression of anthocyanin biosynthetic genes, *PpANS* and *PpUFGT*, by 2.9-fold and 1.9-fold higher, respectively, which correlated well with the color change in the flesh. For transient overexpression assay in the peel, anthocyanin content around the site infiltrated with *PpMYB10.1*/*PpbHLH3*/*PpABCC1* showed a 1.5-fold increase compared to that around the site infiltrated with *PpMYB10.1*/*PpbHLH3*. The expression of *PpANS* in the peel around the site infiltrated with *PpMYB10.1*/*PpbHLH3*/*PpABCC1* was 1.7-fold higher than that in the peel around the site infiltrated with *PpMYB10.1*/*PpbHLH3*, but no significant difference was observed in the expression of *PpUFGT* (Figure 7C). These results suggested the role of *PpABCC1* in promoting anthocyanin accumulation in peach fruit. 

The VIGS system was used to further validate the role of *PpABCC1* in anthocyanin accumulation in peach fruit. *Agrobacterium* cultures containing pTRV1/pTRV2 or pTRV1/pTRV2-*PpABCC1* constructs were transiently infiltrated into the fruits of blood-fleshed peach ‘Dahongpao’ at the color break stage. After infiltration for a week, a decreased pigmentation was observed in the flesh tissues around the site infiltrated with pTRV1/pTRV2-*PpABCC1* as compared to the flesh tissues around the site infiltrated with pTRV1/pTRV2 (Figure 8A). Consistently, anthocyanin content (0.039 mg/100 g FW) in the flesh around the site infiltrated with pTRV1/pTRV2-*PpABCC1* was significantly lower than that in the flesh around the site infiltrated with pTRV1/pTRV2 (0.064 mg/100 g FW). Silencing of the *PpABCC1* resulted in a significant decrease in the expression of *PpANS* (Figure 8A). These results demonstrated that silencing of *PpABCC1* reduced anthocyanin accumulation in peach fruit. 

### 2.9. Transcriptional Activation of PpABCC1 by PpMYB10.1

To verify whether the anthocyanin-related TFs had the ability to regulate the promoter of *PpABCC1*, Y1H and dual luciferase assays were carried out. Since *PpMYB10.1* has been previously reported to be the master regulator of anthocyanin accumulation in peach fruits [38,39], we investigated whether it could activate the transcription of *PpABCC1*. Y1H assay indicated that yeast cells harboring the *PpMYB10.1* gene and the *PpABCC1* promoter could grow on SD/Ura-Leu medium containing 50 ng/mL AbA (Figure 8B). Dual luciferase assay further showed that *PpMYB10.1* had a strong activation activity on the promoter of *PpABCC1* (Figure 8C). These results suggested that *PpMYB10.1* could activate the transcription of *PpABCC1*. 

## 3. Discussion

ABC proteins constitute one of the largest and most diverse transporter families ubiquitously present in different species. ABCs are involved in the transport of a wide range of substrates, such as secondary metabolites, phytohormone transport across membranes and heavy metal detoxification [1,2,3]. In this study, we conducted a comprehensive genome-wide and functional analysis of the ABC transporter gene family in order to determine its role in anthocyanin accumulation in peach fruit. A total of 132 *PpABC* genes were identified in the peach genome, which was divided into eight subfamilies and unevenly located on eight chromosomes.

Gene diversification and evolution caused variations in the chemical characterization of the *PpABC* transporter family. Sequence analysis revealed that the *PpABC* transporter family had notable differences in length, Mw, theoretical pI, total number of negatively/positively charged amino acids, GRAVY, aliphatic and instability index. Subcellular localization showed a majority of *PpABC* (103) genes were localized in the plasma membrane. The exon–intron structural diversity is essential in the evolution of gene families and supports the phylogenetic classification of genes in an organism [40,41]. In this study, closely related members showed similarities in structure and alignment. A typical full-size ABC protein has over 1200 amino acids [9]. The size of the 132 *PpABC* proteins ranged from 71 to 2524 amino acids. The number of exons in *PpABC* genes ranged from one to forty, which was largely consistent with the gene structure in *Arabidopsis* [16] and strawberry [8]. Motif structural analysis revealed a total of fifteen conserved motifs among the different subfamilies. A similar motif sequence pattern was also observed in closely related members indicating that these ABC proteins might perform similar functions.

Whole genome duplication, segmental duplication and tandem duplication mechanisms are the major causes of rapid gene family expansion and evolution in plants [42]. Previous studies have identified duplication events in the ABC genes in different lineages indicating the unending process of gene evolution [43]. Gene duplication events caused a rapid expansion of ABC genes in different species [44,45]. Here, our results showed that ABCB, ABCC, ABCG and ABCI expanded more than other subgroups in peach, thus confirming the diverse nature of the ABC transporters genes and their potential for rapid gene expansion and evolution from their progenitors. To further understand the evolutionary and collinearity relationship of the *PpABC* gene family, we performed the syntenic collinearity analysis of the ABC family in peach. The results suggested that ABC syntenic gene pairs mainly occurred in the ABCG subgroup, indicating a possible evolutionary relationship among the subgroup members.

Expression pattern analysis of *PpABC* family genes was conducted in different tissues and fruits to investigate the function of the ABC genes in peach. Expression profiles in different tissues help to improve our understanding of the tissue-specific and dynamic molecular function of genes. The preferential expression patterns of *PpABC* genes suggested specificities for certain tissues. The majority of the *PpABC* genes displayed high expression levels in the leaf, and low expression levels were detected in the callus (Figure 3). The majority of the ABC genes, including *PpABCC1,* were expressed in all four tissues tested, suggesting their roles in the development of the vegetative and reproductive tissues in peach. Of the *PpABC* genes, 42, 29, 27 and 22 exhibited tissue-specific expression patterns in leaf, callus, root and stem, respectively, suggesting functional divergence of the *PpABC* gene family in peach. The analysis of the fruit expression pattern revealed a high transcript expression level of *PpABC* genes in the ‘Dahongpao’ cultivar. The highest expression levels of six *PpABCC* gene members were observed at the S2 fruit stage, implying that these genes are potentially related to fruit ripening. However, functional redundancies between members are to be expected due to the presence of several *PpABCC* genes and their differential expression levels during fruit development. These results provide a baseline for the identification of various functional genes in peach. 

Anthocyanin is synthesized in the cytosolic surface of the endoplasmic reticulum and then transported into the vacuole for storage. Although intracellular transport is also an important rate-limiting step for anthocyanin accumulation, the related mechanism remains poorly understood. Over the last few decades, different types of tonoplast-localized transporters, such as MATE and ABCC, have been unraveled to participate in anthocyanin accumulation. The involvement of ABCC members in trafficking anthocyanins was found in several plants. In this study, a peach ABCC gene (*PpABCC1*), the ortholog of *Arabidopsis AtABCC2*, a well-known gene for flavonoid transport [26], was functionally characterized to participate in the vacuolar accumulation of anthocyanins. *PpABCC1* was differentially expressed in ‘Dahongpao’ and ‘Jinxiang’ at developmental stages. Additionally, the expression of *PpABCC1* showed a significant correlation with anthocyanin content in ripe fruit of different peach cultivars, indicating that it might be the major ABC transporter in anthocyanin accumulation. Transient silencing of the *PpABCC1* gene caused a significant decrease in anthocyanin accumulation in peach fruit, while its transient overexpression resulted in a deeper coloration in peach fruit. Taken together, these results undoubtedly indicated the essential role of *PpABCC1* in the anthocyanin pigmentation of fruits in peach. It is worth noting that the *PpABCC1* gene was expressed in all four tissues examined in this study. Thus, it is reasonable to speculate that the *PpABCC1* gene plays an important role in anthocyanin accumulation in both vegetative and reproductive tissues. 

Numerous discoveries have revealed that anthocyanin biosynthesis is regulated by the MYB-bHLH-WDR (MBW) transcriptional complex, which is highly conserved in a variety of plant species. In peach, *PpMYB10.1* plays a critical role in the regulation of anthocyanin accumulation. PpMYB10.1 has been shown to interact with PpbHLH3 to induce anthocyanin biosynthesis in blood-fleshed peach [38,46]. Analysis of *cis*-elements indicated the presence of an MYB binding site element (MBSI) in the promoter region of *PpABCC1.* Therefore, we speculated that *PpABCC1* might be regulated by the master regulator PpMYB10.1, such as anthocyanin biosynthetic genes. The Y1H and dual luciferase assay indicated the transcriptional activity of *PpABCC1* was up-regulated by *PpMYB10.1*. Our results found that *PpMYB10.1* could trans-activate the expression of *PpABCC1*, suggesting that *PpMYB10.1* was a key transcription factor in anthocyanin accumulation due to its dual role in anthocyanin biosynthesis and transport. Moreover, light-responsive elements, low temperature-responsive elements and hormone-responsive elements (MeJA-responsive elements, gibberellin-responsive elements, auxin-responsive elements) were found in the promoter of *PpABCC1* (Figure 5). Thus, the expression of *PpABCC1* may be regulated by various external environments and physiological factors such as light, temperature and hormones. Given that anthocyanin accumulation is influenced by various abiotic and biotic stresses, it is possible that *PpABCC1* implicated in anthocyanin transport is sensitive to internal and external stimuli.

Anthocyanin accumulation is a complex process that is determined by synthesis, transport and degradation. Efforts have been made in recent years to understand the mechanism underlying the intracellular transport of anthocyanin in fruit. In a previous study, loss-of-function mutation of a GST gene has been shown to affect anthocyanin accumulation in peach [47]. However, how anthocyanins are transported into the vacuolar lumen by membrane transporters is still not clear. In this study, an ABC membrane transporter was found to participate in anthocyanin accumulation in peach fruit. Our results provide valuable information for further research into the molecular mechanisms underlying anthocyanin transport in fruit.

## 4. Materials and Methods

### 4.1. Plant Materials

All peach accessions used in this study are maintained at the Wuhan Botanical Garden of the Chinese Academy of Sciences, Wuhan, China. Two peach cultivars, a blood-fleshed peach ‘Dahongpao’ and yellow-fleshed peach ‘Jinxiang’, were selected for gene expression profiling and RNA-seq analysis. Fruit samples were collected at two developmental stages, green stage (S1) at 65 days after full blossom (DAFB) and color break stage (S2) at 85 DAFB. Flesh samples were cut into small pieces, immediately frozen in liquid nitrogen and stored at −80 °C until use. For each sample, three biological replicates were conducted, with at least four fruits in each replicate. Additionally, peach fruits used for transient overexpression and virus-induced gene silencing (VIGS) assays were collected at the color break stage. 

### 4.2. Identification and Sequence Analysis of the ABC Gene Family in Peach

Peach ABCs were identified by running a basic local alignment search tool (BLAST) search against the protein sequence database of peach (https://www.rosaceae.org/; accessed on 4 March 2022) using *Arabidopsis* ABC protein sequences as query sequences. The length, molecular weight (Mw), theoretical pI, total number of negatively/positively charged amino acids, grand average of hydropathicity (GRAVY), and aliphatic and instability index of the peach ABC family were calculated by ExPASy (https://web.expasy.org/protparam/; accessed on 10 May 2022). Subcellular localization was predicted by CELLO (http://cello.life.nctu.edu.tw/; accessed on 16 May 2022). Chromosomal distribution of the ABC transporter genes was visualized using MapChart [48]. Advanced Circos from TBtools software [49] was used to perform a collinearity analysis on the ABC genes to identify the duplicated genes. All parameters were set to the software’s default values.

### 4.3. Phylogenetic, Conserved Motifs and Gene Structure Analysis

Sequence alignment of the ABC protein sequences in peach was performed using ClustalX (1.81). The phylogenetic tree was constructed using the neighbor-joining algorithm in MEGA 6.0 with 1000 bootstrap replicates [50]. The conserved motifs of the putative peach ABC proteins were determined using the online MEME suite (http://meme-suite.org/tools/meme; accessed on 28 May 2022). The MEME results were visualized using TBtools. Gene structural analysis of the exon–intron organization was identified using the online Gene Structure Display Server (GSDS 2.0; http://gsds.cbi.pku.edu.cn; accessed on 2 June 2022) program [51]. All parameters were set as the default value of the software.

### 4.4. Cis-Acting Regulatory Element Analysis

To predict putative *cis*-acting elements in the ABC family genes, the promoter sequences 1.5 kb upstream of the ATG start codon were extracted from peach genome. The *cis*-acting elements were then predicted using the online Plant Cis-acting Regulatory DNA Elements (PlantCARE) database (http://bioinformatics.psb.ugent.be/webtools/plantcare/html/; accessed on 20 May 2022). 

### 4.5. RNA Extraction and Quantitative Real-Time PCR

Total RNA was extracted from peach fruit samples using the Total RNA Rapid Extraction Kit (Magen, Guangzhou, China). Removal of genomic DNA and first strand cDNA synthesis were carried out using PrimeScript^TM^ RT reagent Kit with gDNA Eraser (Takara, Dalian, China). Quantitative real-time PCR (qRT-PCR) was conducted using SYBR^®^ Premix Ex Taq™ II (Takara, Dalian, China), with the following amplification program: one cycle of 30 s at 95 °C, followed by 40 cycles of 5 s at 95 °C and 30 s at 60 °C. The peach *PpTEF2* gene encoding a translation elongation factor was used as an internal control to normalize the expression of all target genes [52]. All analyses were repeated using three biological replicates. The quantification of gene expression level was calculated using the 2^−ΔΔCT^ method [53]. Primer sequences used for gene expression are listed in Appendix A. 

### 4.6. Transcriptome Data Analysis

The transcriptome data of two cultivars, ‘Dahongpao’ and ‘Jinxiang’, were used to investigate expression profiles of the *PpABC* genes at two developmental stages. The RNA-seq data were extracted from our previous study [38]. Gene expression levels were estimated based on the expected number of fragments per kilobase of transcript sequence per millions of base pairs sequenced (FPKM). DESEQ2 was used to identify differentially expressed genes (DEGs) [54], with the following parameters: log_2_ fold change (FC) cut off of ±1 and a false discovery rate (FDR) ≤ 0.05. In addition, expression patterns of the *PpABC* genes in the leaf, callus (induced from cotyledon), root and stem tissues were investigated using the transcriptome data of the peach cultivar ‘Zaoyoutao’. Gene expression levels were estimated based on the FPKM values, which were used to generate a heatmap using TBtools. 

### 4.7. Extraction and Quantification of Anthocyanin Content

Extraction and quantification of anthocyanin content were carried out using the Plant Anthocyanin Assay Kit (Solarbio, Beijing, China). The absorbance of the samples was determined at 530 and 700 nm using an Infinite M200 luminometer (Tecan, Mannerdorf, Switzerland). Total anthocyanin (TA) content was calculated according to a previous report [55]. Three biological replicates were performed for each sample. 

### 4.8. Yeast One-Hybrid Assay (Y1H)

The yeast one-hybrid assay (Y1H) was conducted using the Matchmaker^®^ Gold Yeast One-Hybrid Library Screening System User Manual (Clontech, Palo Alto, CA, USA) kit. The promoter sequence of *PpABCC1*, 2 kb upstream of the start codon, was amplified and inserted into the reporter vector pAbAi to generate pBait-pAbAis constructs. The full-length coding sequence of the *PpMYB10.1* was inserted into the effector plasmid of pGADT7 to generate the pGADT7-*PpMYB10.1* construct. Positive yeast cells were used to determine the activity of the pGADT7-TF with pBait-pAbAis on SD/-Ura/AbA* medium (*, the minimal inhibitory concentration of AbA). Primer sequences used for Y1H are listed in Appendix A.

### 4.9. Dual Luciferase Reporter Assay

Dual-luciferase reporter assay in *Nicotiana benthamiana* leaves was conducted according to a previously described protocol [39]. Promoter sequences 2 kb upstream of the ATG start codon were inserted into the pGreen II 0800-LUC vector. The constructs were individually transformed into the *Agrobacterium* strain GV3101 containing the pSoup helper plasmid. *Agrobacterium* cultures carrying the constructs were then resuspended in infiltration buffer (10 mM MES, 10 mM MgCl_2_, 200 μM acetosyringone, pH 5.7) to an optimal density (A_600_ = 0.75) and incubated at 25 °C without shaking for 2 h before infiltration. The mixture of *Agrobacterium* containing the transcription factor (1 mL) and the promoter (100 μL) was injected into *N. benthamiana* leaves. After three days of infiltration, the ratio of firefly luciferase (LUC) to renilla luciferase (REN) activity was measured using the Dual-Glo^®^ Luciferase Assay System (Promega, MI, USA). Four biological replicates were performed for each treatment. Primer sequences used for vector construction are listed in Appendix A.

### 4.10. Subcellular Localization Assay

Subcellular localization was examined by transiently expressing the *PpABCC1*-YFP in *N. benthamiana* leaves using *Agrobacterium* (GV3101), similar to the protocol described for the dual-luciferase assay above. Tonoplast marker and transgenic *N. benthamiana* plants containing a red fluorescent nuclear marker (Nucleus-RFP) were used for the identification of the subcellular structures [56]. Fluorescence was detected three days after infiltration using the confocal microscope (TCS SP8, Leica, Microsystems, Wetzlar, Germany). Primer sequences used for vector construction are listed in Appendix A.

### 4.11. Transient Overexpression and Virus-Induced Gene Silencing (VIGS) in Tobacco and Peach Fruit

Transient overexpression was carried out by inserting full-length coding sequences of *PpABCC1* into the pSAK277 vector, which was then transformed into the *Agrobacterium* strain GV3101. *Agrobacterium* cultures were then resuspended in an infiltration buffer using the same protocol as described above for the dual-luciferase assay. The infiltrated tobacco (*Nicotiana tabacum*) leaves and peach fruits of ‘Huangjinmi’ were placed in a growth chamber (25 °C under 16 h light and 8 h dark photoperiod). Five days after infiltration, digital photographs were taken. For the VIGS system, *Agrobacterium* cultures containing pTRV1 and pTRV2 or pTRV2-*PpABCC1* were mixed in a ratio of 1:1 and infiltrated into the blood-fleshed peach fruit. The injected fruits were placed in a growth chamber (25 °C under 16 h light and 8 h dark photoperiod). Digital photographs were taken one week after infiltration. Each treatment was performed with three biological replicates with at least ten fruits per replicate. Primer sequences used for vector construction are listed in Appendix A.

## Figures and Tables

**Figure 1 ijms-24-01931-f001:**
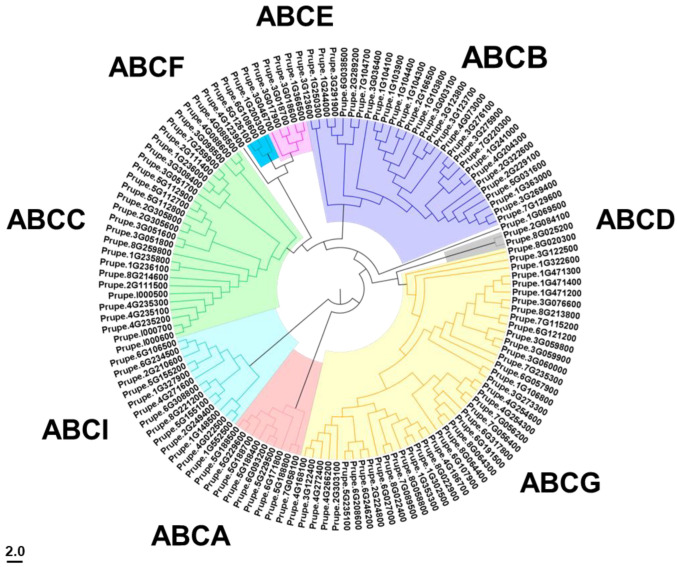
Phylogenetic analysis of ABC proteins from peach. The phylogenetic tree was constructed using the neighbor-joining method in MEGA 6.0 with 1000 bootstrap replicates. Different colors highlighted the eight subfamilies.

**Figure 2 ijms-24-01931-f002:**
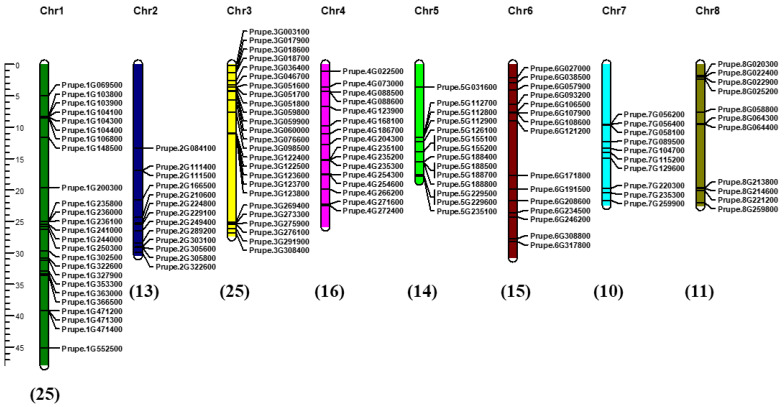
Chromosomal distribution of *PpABC* genes. The chromosomes were visualized using MapChart program. The chromosome number is indicated at the top of each chromosome, and the number of *PpABC* genes in each chromosome is indicated at the bottom of each chromosome in brackets. The scale bar described the relative lengths of the chromosomes in megabases (Mb).

**Figure 3 ijms-24-01931-f003:**
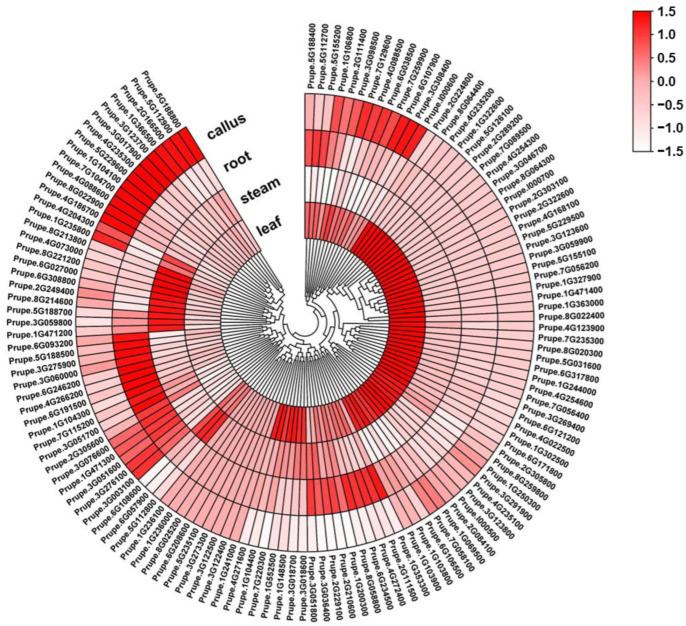
Expression profile of *PpABC* genes in different tissues of peach. The FPKM values were used to generate a heatmap with hierarchical clustering analysis. The color scale displaying the expression levels (−1.5 to 1.5) was indicated.

**Figure 4 ijms-24-01931-f004:**
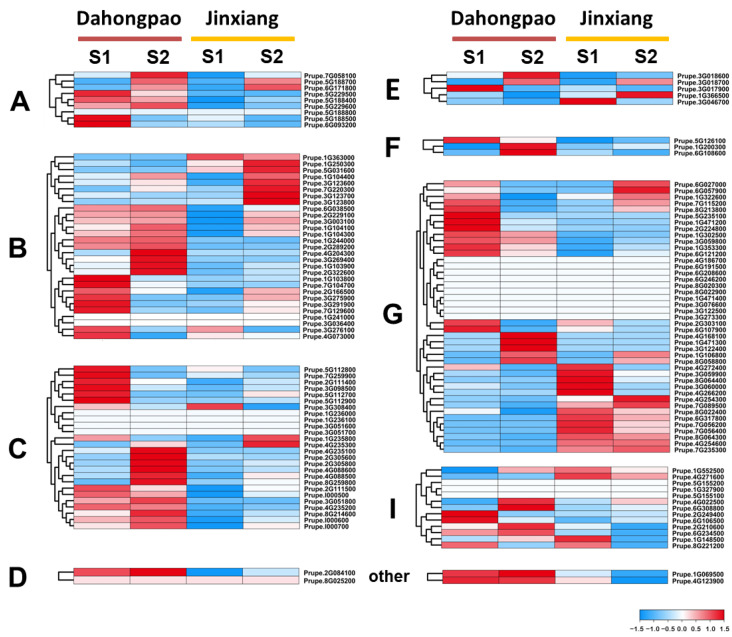
Expression profile of *PpABC* genes during fruit development of peach. Transcript expression levels were determined during two developmental stages, S1 (green stage at 65 DAFB) and S2 (color break stage at 85 DAFB). The FPKM values were used to generate a heatmap with hierarchical clustering analysis. A–G and I indicated the eight subfamilies of *PpABC* genes. The color scale displaying the expression levels (−1.5 to 1.5) was indicated.

**Figure 5 ijms-24-01931-f005:**
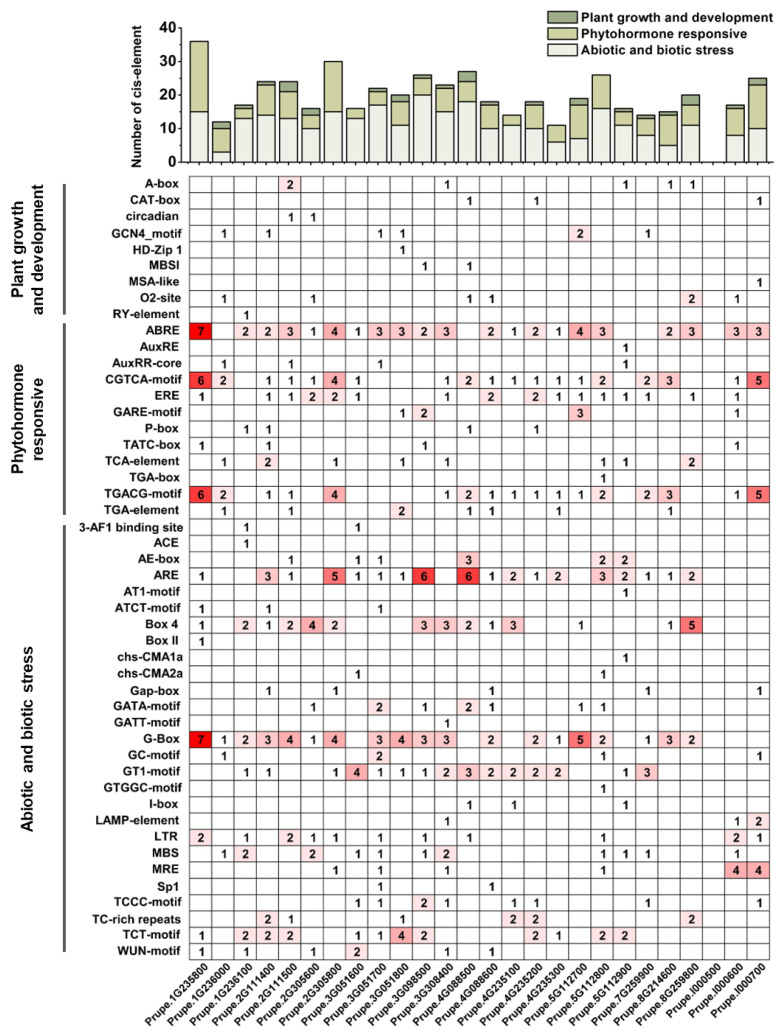
*Cis*-elements in the 1.5-kb upstream region of *PpABCC* genes. The bars on the top represent the total number of *cis*-elements in each gene promoter region. Different colors indicate different types of *cis*-elements. The color intensity and number in the cells indicated the number of *cis*-element in these *PpABCC* genes.

**Figure 6 ijms-24-01931-f006:**
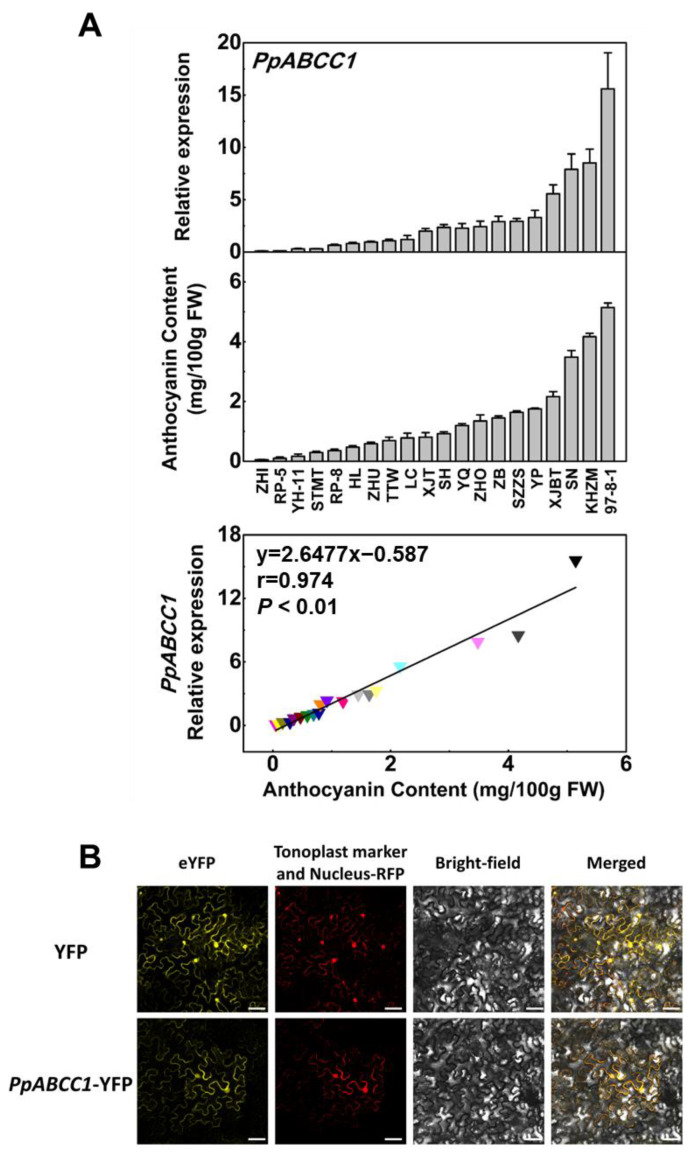
Expression and subcellular location for the *PpABCC1* gene: (**A**) expression of *PpABCC1* and anthocyanin content in ripe fruits of 20 peach cultivars and the correlation analysis between the *PpABCC1* expression and anthocyanin content. Data represent means (±SD) of three biological replicates; (**B**) subcellular localization assay of PpABCC1 in tobacco leaves. Scale bars indicated 50 μm.

**Figure 7 ijms-24-01931-f007:**
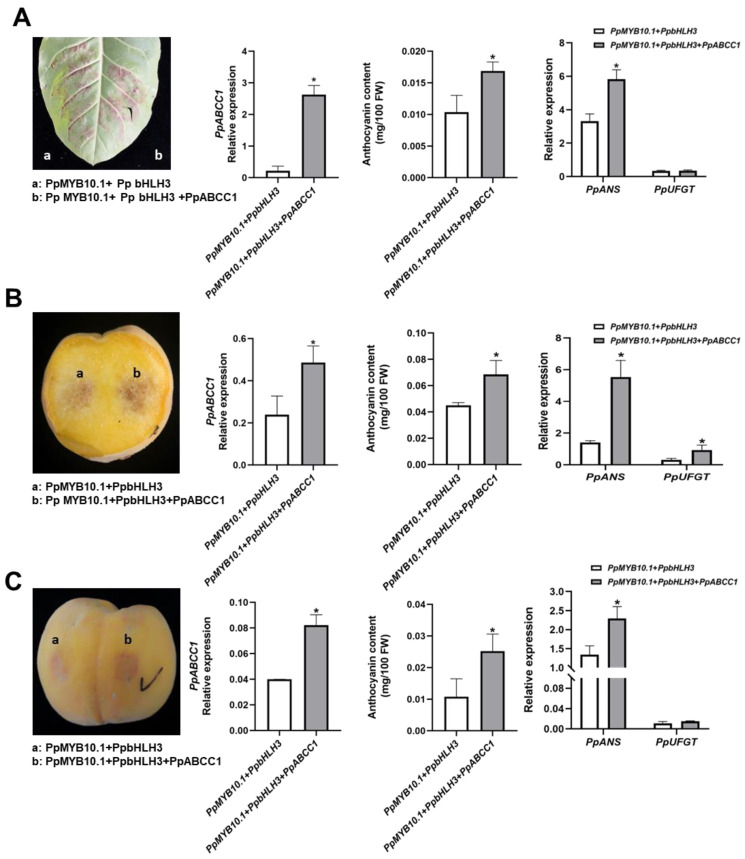
Transient overexpression of *PpMYB10.1* + *PpbHLH3* with or without PpABCC1 in tobacco leaves (**A**) and peach (cv. ‘Huangjinmi’) fruit (**B**,**C**). Relative expression of *PpABCC1* and anthocyanin biosynthetic genes, changes in anthocyanin contents in tobacco leaves (**A**) and peach fruit (**B**,**C**). Data were means (±SD) from three biological replicates. *, Significant difference at *p* < 0.05 level.

**Figure 8 ijms-24-01931-f008:**
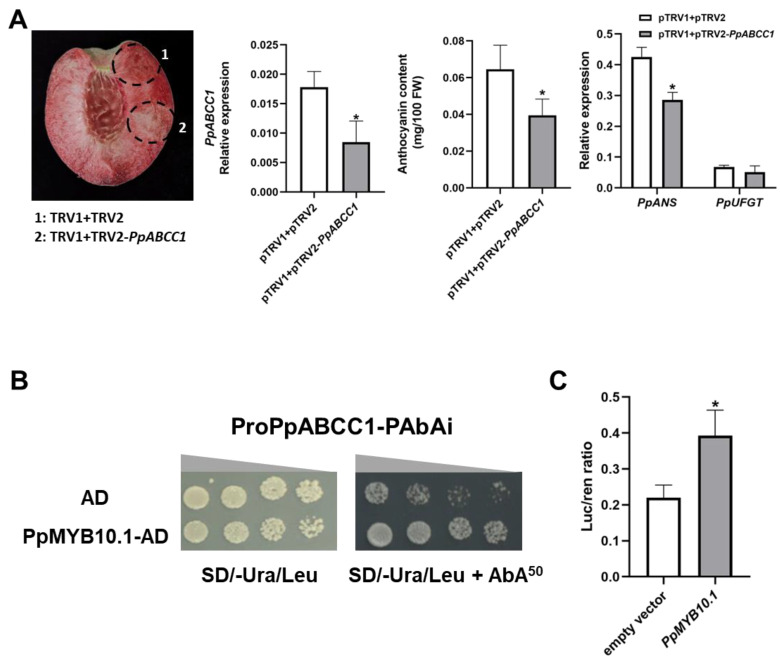
Functional analysis of the *PpABCC1* gene: (**A**) Transient silencing of *PpABCC1* in the fruit of blood peach ‘Dahongpao’ at S2 using virus-induced gene silencing (VIGS) system. Expression of *PpABCC1* and anthocyanin biosynthetic genes and anthocyanin content were measured in the flesh tissues around the infiltration sites. (**B**) Assay of the interaction between PpMYB10.1 and the promoter of *PpABCC1* using yeast one-hybrid (Y1H). (**C**) Assay of the activation activity of PpMYB10.1 on the promoter of *PpABCC1* using dual luciferase reporter system. Data represented means (±SD) of three independent biological replicates. *, Significant difference at *p* < 0.05 level.

## Data Availability

Not applicable.

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
