# Peer review of "Genome-Wide Analysis of ATP Binding Cassette (ABC) Transporters in Peach (Prunus persica) and Identification of a Gene PpABCC1 Involved in Anthocyanin Accumulation"

_ijms, 2023, doi:10.3390/ijms24031931_

Round 1

Reviewer 1 Report

Dear Authors

The experiment is very well designed and covering all the aspects about ABC gene families. Writing style and English language is good. Some minor suggestions:

Please be careful about the abbreviations, some of them doesn't have the full explanations for instance in the line 43 what are the TMD and NBD?

Discussion part is too short and needs to be written better. 

Best wishes

Author Response

#Reviewer 1

The experiment is very well designed and covering all the aspects about ABC gene families. Writing style and English language is good.

Response: We appreciate the reviewer’s comments.

Some minor suggestions:

Comment 1: Please be careful about the abbreviations, some of them doesn't have the full explanations for instance in the line 43 what are the TMD and NBD?

Response: The full names for the abbreviations of TMD and NBD were provided in the main text. Please check lines 53 and 54.

Comment 2: Discussion part is too short and needs to be written better.

Response: Thank you for the instructive comment. We have substantially revised the discussion section. 

Reviewer 2 Report

The reviewed paper reports the results of a big cross-examination of a big gene family in a non-model plant species with a special focus on gene(s) controlling anthocyanin accumulation in a fruit. The approaches are valid and the results are more or less clearly presented. The whole text is well written. I have no doubts this paper deserves publication in IJMS.
However, I have two major concerns about this paper (not the work itself). Both are easy to correct.
1. Some improvement is required in the statistical part of this work. SEs should be replaced with SDs, as SEs are not good indicators of variation. For explanation why, I recommended a paper cited in a note directly in a manuscript file (see attached). Figure 6A should be supplemented by the correlation coefficient.
2. The Discussion section should be improved. I recommend to move all concluding remarks from the Results section there. Otherwise states, the Results section are recommended to report the results themselves, while the Discussion section should focus on what these results may mean. There should be no repeats of the introductory section in the Discussion.
There are additional notes and comments in a manuscript file which are, I believe, relatively easy to deal with. In its updated form this valuable research can be published in IJMS.

Author Response

#Reviewer 2

The reviewed paper reports the results of a big cross-examination of a big gene family in a non-model plant species with a special focus on gene(s) controlling anthocyanin accumulation in a fruit. The approaches are valid and the results are more or less clearly presented. The whole text is well written. I have no doubts this paper deserves publication in IJMS.

Response: We appreciate the reviewer’s comments.

However, I have two major concerns about this paper (not the work itself). Both are easy to correct.

Comment 1: Some improvement is required in the statistical part of this work. SEs should be replaced with SDs, as SEs are not good indicators of variation. For explanation why, I recommended a paper cited in a note directly in a manuscript file (see attached). Figure 6A should be supplemented by the correlation coefficient.

Response: Thank you for the instructive comment. We have changed ‘SEs’ to ‘SDs’ in all figures as suggested.

Comment 2: The Discussion section should be improved. I recommend to move all concluding remarks from the Results section there. Otherwise states, the Results section are recommended to report the results themselves, while the Discussion section should focus on what these results may mean. There should be no repeats of the introductory section in the Discussion.

Response: Thanks for these useful comments and suggestions. We have substantially revised the discussion section.

There are additional notes and comments in a manuscript file which are, I believe, relatively easy to deal with. In its updated form this valuable research can be published in IJMS.

Response: We corrected all the mistakes according to the reviewer’s comments. And the details are listed as follows.

Line 3: We have added the ‘Prunus persica’ and ‘gene’ in the title.

Line 32: We have formatted citations according to IJMS requirements in the revised manuscript.

Line 42: The word ‘arrangements’ was replaced with ‘classified into three types based on their domain structure’ (see lines 58-59).

Line 50: The word ‘of’ was removed. And ‘full sized’ was edited to ‘full-sized’ (see line 65).

Line 50-80: The whole paragraph was revised as suggested. We have added a table containing the information of ABC family (Table S1).

Line 85: The abbreviation (GSH) was explained at its first mention (see line 87).

Line 96: The word ‘crop’ was replaced by ‘plant species’ (see line 108).

Line 101: The letter ‘e’ was added to the word ‘Rosaceae’ (see line 111).

Line 110: Revised as suggested. Please check lines 119-121.

Line 125: We sampled the fruits from different several trees of the same cultivar.

Line 131: The word ‘Arabidopsis’ was italicized (see line 443).

Line 157: The word ‘contamination’ was removed (see line 472).

Line 175: The sentence was rephrased for clarity and the word ‘significant’ was removed (see line 489-490).

Line 181: The name of the spectrophotometer used was provided (see line 499-501).

Line 183: We cited a previously published work where this formula was used (see line 501).

Line 202: ‘A600 = 0.75’ was replaced to ‘A600 = 0.75’ (see line 521).

Line 253: The word ‘space’ was added after the word ‘extracellular’ (see line 145).

Line 259: The ABCG family had the largest number of members. We have rephrased for clarity. The sentence has been changed to ‘largest number of members in all subfamilies’ (see line 152).

Line 264: Revised as suggested, the scale bar was on the left (see figure file, Figure 1). ‘With 1000 bootstrap replicates’ was added to the figure legends (see line 567).

Line 273: We have rephrased for clarity and the sentence was changed to ‘In peach, the numbers of the ABCB, ABCC, ABCG and ABCI subfamilies were much higher than those of the other subfamilies’ (see line 161-163).

Line 276: The word ‘gene’ was removed (see line 166).

Line 278-279: The sentence was deleted (see line 168-169).

Line 279: ‘forty’ was added after exon (see line 168).

Line 295: Rephrased for clarity. We have shortened this part (see line 185-192).

Line 311: The sentence has been rephrased for clarity ‘These results suggested that segmental duplications contributed to the expansion of PpABC genes’ (see line 196).

Line 315: ‘in brackets’ was added after ‘chromosome’ (see line 573).

Line 319: The callus is induced from cotyledon; we have added this in the materials and methods (491).

Line 321-326: The percentage values were edited (see lines 203-207).

Line 323: Revised as suggested. We have removed this sentence.

Line 323: ‘levels’ was added to the word ‘expression’ (see line 204).

Line 325: We have added ‘of 123 expressed PpABC genes’ in this sentence (see line 206).

Line 328: Revised as suggested. We have removed this sentence.

Line 340: The word ‘had transcription’ was replaced by ‘were expressed’ (see line 216).

Line 343: Revised as suggested. We have removed this sentence.

Line 350: A-I indicated the eight subfamilies of PpABC genes. We have added this in figure legends (Figure 4) (see line 583-584).

Line 356: Noted, all the gene names have been italicized.

Line 363: The sentence has been rephrased to ‘the cis-elements related to abiotic and biotic stress were abundant, including light-responsiveness elements (G-Box) and low temperature-responsive elements (LTR)’ (see line 234-235).

Line 369: Thank you for the instructive comment.

Line 392: Revised as suggested. We have added a list of the twenty peach cultivars used in Table S8.

Line 403: Revised as suggested. We have changed ‘R2’ with the Spearman’s correlation coefficient in figure 6.

Line 404: Revised as suggested. We have replaced all (SEs) with (SDs) in figure 6.

Line 405: Revised as suggested. We have provided larger photos with clear scale bars (see figure file, figure 6).

Line 410: Revised and italicized gene names in the whole manuscript.

Line 425: Revised as suggested. We have replaced all (SEs) with (SDs) in figure 7.

Line 432: PpANS and PpUFGT are the most important structural genes in anthocyanin biosynthetic pathway. We have added ‘anthocyanin biosynthetic genes’ in this sentence (see line 289).

Line 454: Revised as suggested. We have changed the peach photo (see figure file, figure 8).

Line 460: Revised as suggested. We have replaced all (SEs) with (SDs) in figure 8.

Line 471: Revised the whole discussion section.

Line 475: The paragraph was omitted as suggested (see line 327).

Line 555: Revised and formatted the reference section accordingly.